# Clinical Effectiveness of Renal Transplant Outpatient Pharmaceutical Care Services in Korea

**DOI:** 10.3390/healthcare11182597

**Published:** 2023-09-21

**Authors:** Ha Young Jang, Yon Su Kim, Jung Mi Oh

**Affiliations:** 1College of Pharmacy, Gachon University, Incheon 21936, Republic of Korea; hyjang@gachon.ac.kr; 2College of Pharmacy, Research Institute of Pharmaceutical Sciences, Seoul National University, Seoul 08826, Republic of Korea; 3Department of Internal Medicine, Seoul National University Hospital, Seoul 03080, Republic of Korea; yskim@snu.ac.kr; 4Department of Internal Medicine, Seoul National University College of Medicine, Seoul 03080, Republic of Korea

**Keywords:** pharmaceutical care service, kidney transplant, outpatient clinic, clinical effectiveness

## Abstract

Background: The necessity and importance of pharmaceutical care services (PCS) are well recognized, yet the concept and scope of PCS have not yet been clearly defined in Korea, particularly in kidney transplantation outpatient clinics. Aim: The main purpose of this study is to evaluate whether PCS is effective in the outpatient setting for kidney transplant patients. Methods: For three years, a clinical pharmacist provided PCS to kidney transplant patients in an outpatient setting to evaluate the clinical effectiveness of PCS. Results: A total of 302 patients were matched in a 1:1 ratio, with 151 in the PCS group and 151 in the control group. These patients were followed, and a total of 476 interventions were provided to them, including medication reconciliation (n = 113, 23.7%), medication evaluation and management (n = 186, 39.1%), and pharmaceutical care transition (n = 177, 37.2%) services. The estimated glomerular filtration rate (eGFR) exhibited a notable difference between the control and PCS groups when comparing the pre- and post-study periods measurements. In the control group, there was a decline of 7.0 mL/min/1.73 m^2^ in eGFR. In contrast, the PCS group showed a smaller decline of 2.5 mL/min/1.73 m^2^ (*p* = 0.03). The adjusted odds ratio for end stage renal disease development in the PCS group was 0.51 (95% confidence interval: 0.26–0.96), indicating a significantly lower risk compared to the control group. Conclusion: Our study highlights the promising potential of PCS implementation in kidney transplantation outpatient clinics. Further research is needed to validate and expand upon these findings, especially in diverse clinical settings.

## 1. Introduction

Pharmaceutical care services (PCS) are patient-centered services aiming to increase the safety and effectiveness of drug treatments and improve the patients’ quality of life [1,2]. The ability of PCS provided by pharmacists to improve the clinical, economic, and humanistic outcomes of patients has been well established [3]. Services such as medical history management, drug treatment plan management, adverse drug reaction management, drug information education, and medication counseling by a pharmacist are effective in reducing adverse drug reactions [4,5,6]. PCS implementation significantly improved human performance indicators, including medication adherence, knowledge about diseases and drugs, and health-related quality of life of the patient [7]. PCS also improved the overall quality of life of the patients by relieving anxiety related to their disease and social life.

Among various medical specialties, PCS for renal transplantation patients is an area in need of development because patient adherence to medications, including immunosuppressants, is crucial to maintain the patient’s graft function [8]. In the USA, the United Network for Organ Sharing (UNOS), which administers the organ procurement and transplantation network [9], requires all transplant programs to identify at least one pharmacist responsible for providing pharmaceutical services to solid organ transplant recipients. The American Society of Health-System Pharmacists (ASHP) also emphasizes the roles of pharmacists as members of a multidisciplinary team in all three transplantation phases (pre-transplantation, perioperative, and post-transplantation) [10]. The role of transplant pharmacists has steadily increased over time, with a dedicated focus on pharmacological evaluation, patient education, and medication coverage. These specialized pharmacists play a crucial role in ensuring optimal medication management for transplant patients, addressing their specific needs, and enhancing the overall quality of care provided in transplantation settings. Given the necessity and importance of PCS, efforts have been made in South Korea to develop and standardize these pharmaceutical services. However, these attempts have encountered two major limitations. Firstly, there is insufficient evidence regarding the effectiveness of PCS in Korea. As a result, PCS is currently not covered by insurance or supported by the government in Korea. This highlights the need for further research to evaluate and establish the effectiveness of PCS in the Korean healthcare system.

Secondly, providing outpatient PCS in Korea presents specific challenges compared to inpatient settings. One significant challenge is the limited time available for PCS in an outpatient setting. The average duration of an outpatient visit, such as in nephrology clinics, is notably short, often around 3.7 min [11]. This time constraint makes it difficult for pharmacists to thoroughly identify and address drug-related problems (DRP) during these brief encounters. This short time for outpatient PCS necessitates finding innovative approaches to maximize the effectiveness of pharmaceutical care within these limitations.

Another challenge associated with PCS in Korea is the shortage of pharmacists specialized in the areas of transplantation pharmaceutical care. This shortage can hinder the effective implementation of PCS services, as there may not be enough pharmacists available to provide comprehensive patient care and perform necessary medication management tasks. Despite the presence of 353 tertiary hospitals in Korea, the number of pharmacists who specialize in organ transplantations is limited. According to the Korean Society of Health-System Pharmacists, there are only 72 pharmacists with expertise in this field. This shortage of specialized pharmacists further exacerbates the challenges faced in providing effective PCS for patients undergoing kidney transplantations [12,13].

To date, only a limited number of studies have examined the effectiveness of PCS in renal transplant patients. While many studies have assessed the clinical effectiveness of PCS, a few have systematically targeted Asian populations [14], especially Koreans. Most of these studies encompass fewer than 100 patients and are predominantly based in the United States [3,8] or Europe [15]. To address the aforementioned concerns, we implemented clinical PCS in a Korean kidney transplant outpatient clinic for three years. The main purpose of this study is to evaluate whether PCS is effective in the outpatient setting for kidney transplant patients.

## 2. Materials and Methods

### 2.1. Pharmaceutical Care Service (PCS)

This study was conducted in collaboration with the nephrology outpatient clinic at the Seoul National University Hospital (SNUH) in South Korea. The study participants were individuals who visited the clinic between July 2019 and June 2022, and they were followed up during this period. The inclusion criteria encompassed all kidney transplant-related outpatients: patients awaiting a transplant, those who had received one, and kidney donors. Patients with end-stage renal disease (ESRD) on dialysis were excluded from the study. According to the ASHP guidelines, PCS could be adjusted and tailored based on the resources and circumstances of each institution [10]. Therefore, we applied the DrugTEAM service algorithm that has been developed PCS by our research team [16,17] to the kidney transplantation outpatient clinic. The DrugTEAM service, a term denoting PCS offered by clinical pharmacists within the Department of Nephrology at SNUH, was established more than 10 years ago. Initially designed for inpatients, it has recently expanded its reach to accommodate outpatients as well. The study was carried out over a span of 3 years, during which a clinical pharmacist (CP) and a nephrologist collaborated closely. The CP was the provider of PCS for patients and healthcare professionals [16]. The CP resided at the outpatient clinic and employed a systematical approach to identify and assess every kidney transplant recipient followed at the clinic by screening their recent medical history and prescription records. The pharmacist examined the patients’ medical charts a day prior to the appointment in preparation for the intervention, spending an estimated three minutes per patient. During the appointment, the pharmacist consulted with the physician concerning the patient. Subsequently, in a separate room, the pharmacist devoted three minutes to furnish the patient with medication guidance. On average, the pharmacist’s engagement with each patient totaled approximately 10 min. The CP involved in the study had extensive experience of five years in dealing with chronic kidney disease (CKD) and kidney transplantation. The CP conducted a structured, patient-centered medication review and actively engaged with both healthcare professionals and patients in a kidney transplantation outpatient clinic. The PCS provided by the CP encompassed medication reconciliation (MR), medication evaluation and management (MEM), and pharmaceutical care transition (PCT) services (Figure 1). The MR Service provided by the CP involved the adjustment of prescription days for patients who had remaining or insufficient medication. This was achieved through the collection, examination, communication, and documentation of the patient’s medication history. The primary goal of the MR service was to minimize discrepancies in prescribed medication use. In MEM service, the pharmacist conducted the assessment of the physician’s prescription to identify any drug-related problems. This MEM involved a systematic process of finding, assessing, recommending, monitoring, and documenting medication-related problems. Based on the assessment, the pharmacist made recommendations to address any identified drug-related problems to the physician at the clinic. In PCT service steps, the CP provided counselling to the patient and caregiver regarding the discharge medications using the educational materials, including written materials covering medication usage instruction, administration timetables, and medication diaries were utilized to aid in reinforcing patient compliance and health knowledge before and after outpatient clinics.

### 2.2. Study Cohort Design

A baseline examination was performed for the participants at their first visit. For every visit, the PCS was provided when the clinical pharmacist determined that the patient needs a PCS. All participants were retrospectively analyzed through a manual chart review, and cohort groups were classified as the intervened or control group based on whether they had received a PCS or not. The requirement for written informed consent from participants was waived because the analysis was carried out retrospectively. This study was approved by the Institutional Review Board (IRB) of the SNUH (IRB No. SNUH-1511-055-719).

### 2.3. Study Outcomes and Covariates

The primary study outcome was the changes in estimated glomerular filtration rate (eGFR) before and after the 3-year of study period. For the secondary outcome, the incidence rate of progression to ESRD after the study period was also compared between the two groups. The following baseline characteristics with potential influence on the study outcomes were included: age; sex; diagnosis (kidney transplantation, CKD, kidney donor, hypertension, diabetes mellitus, and dyslipidemia); number of visits; number of medications; medications (diuretics, potassium-lowering agents, acidosis, urate-lowering agents, antibiotics, immunosuppressants, and chronic kidney disease-mineral bone disorder (CKD MBD) medications) and laboratory findings (albumin (ALB), blood urea nitrogen (BUN), calcium (Ca), glucose, hemoglobin (Hb), serum creatinine (Scr), total bilirubin (TBIL), uric acid (UA), eGFR, platelet (PLT), white blood cell (WBC), and cholesterol).

### 2.4. Statistical Analysis

Descriptive values were presented as means (SD) for continuous variables and as proportions for categorical variables. To address confounding due to intervention, the propensity-score-matching method was applied. Matching was performed using SAS 9.4 (SAS Institute Inc., Cary, NC, USA) Greedy 5 → 1 Digit Match macro [18]. The propensity score was obtained using logistic regression analysis to predict the intervention or control group from 19 covariates. Distribution of patients’ baseline covariates was evaluated with a standardized difference. A standardized difference of <0.1 was considered indicative of good balance [19] (NCSS-statistical-Software, Ch 123, 2017). To construct the outcome model, Student’s t test was used to assess changes in eGFR after the interventions. Logistic regression was used to estimate the odds ratio (OR) of intervention for progression to ESRD, with 95% confidence interval (CI). Values with *p* < 0.05 were considered statistically significant. A sensitivity analysis was performed to ensure the robustness of the study. Another new study cohort was built using optimal match instead of greedy match for propensity-score-matching to see if the results were the same.

## 3. Results

### 3.1. Patient Characteristics

In total, 620 patients who visited the kidney transplantation outpatient clinic were evaluated (Figure 2).

After removing patients with ESRD, the eligible cohort of 522 individuals was left for further analysis. The patients were divided into an intervention group (case, n = 239) and a non-intervention group (control, n = 283). The proportion of patients who underwent kidney transplantation was higher in the intervention group than in the non-intervention group. Significant differences were observed in all co-medications and ten laboratory findings except cholesterol and platelet level, whereas there were no significant differences in sex and age between two groups (Table 1). The PCS patient group had worse medical conditions in terms of kidney function and took more medication. After the intervened group (n = 151) was matched to the control group (n = 151), the above differences were reduced, and both groups were well balanced. Standardized differenced differences were equal to or lower than 0.1 for all covariates. The mean age of patients (53.7 years; men: 57.6% (n = 302)) was shown.

### 3.2. Types of Intervention

A total of 476 PCS interventions were provided to the patients (MR: n = 113; MEM: n = 186; PCT: n = 177) in the intervened group (Table 2). The intervention group received an average of 2.38 interventions, with a maximum of 15 and a minimum of 1. In MR services, medication discrepancies were adjusted for immunosuppressants, hematopoietic agents, CKD-MBD drugs (e.g., phosphorous-binding agents), and insulin. In MEM services, the drug-related problem with the highest demand from physicians was prescribing patient-required medications (indication without prescription), followed by insurance issues, drug information, diabetes management, and dose adjustment. For PCS services, consultations regarding drug usage and drug changes were the two most frequently requested services from patients, followed by lifestyle guidance, side effects, supplements, adjustment of drug schedules, and drug identification.

CKD-MBD, chronic kidney disease-mineral bone disorder; GFR, glomerular filtration rate; GI, gastrointestinal; MEM, medication evaluation and management service; MR, medication reconciliation service; PCT, pharmaceutical care transition service. The items in each line consist of drug classes that have been provided and the PCS service in detail. In MR service, the pharmacist identifies and corrects unintended medication discrepancies for patients. This is to improve patient adherence to medication. The MEM service checks drug-related problems for physicians in prescribing to the physician for the effectiveness and safety of the medication. Each patients’ DRPs were defined according to Pharmaceutical Care Network Europe (PCNE) DRP classification criteria. In PCT service, the pharmacist provided a medication education to patients.

### 3.3. Association with Study Outcomes

The change in eGFR showed a significant difference between the control and PCS groups, with a decrease of −7.0 mL/min/1.73 m^2^ in the control group and −2.5 mL/min/1.73 m^2^ in the PCS group (*p* value = 0.03) (Figure 3). In the MR, MEM, and PCT groups, the changes in eGFR were −0.90, −1.3, and −4.0, respectively. Both the MR and MEM groups exhibited statistically significant differences when compared to the control group (*p* value < 0.05). A total of 45 out of 302 participants developed ESRD (Table 3). Among the control group, ESRD developed in 29 out of 151 (19.2%) individuals, while in the PCS group, ESRD developed in 16 out of 151 (10.6%). The adjusted odds ratio (aOR) for ESRD development in the PCS group was 0.51 (95% CI: 0.26–0.96), indicating a significantly lower risk compared to the control group.

### 3.4. Sensitivity Analysis

In a sensitivity analysis, 167 PCS and 167 control participants were newly matched, with no significant differences in baseline characteristics between the two groups (Appendix A). The results were consistent with the original findings. The incidence of ESRD in the control group was 17.9% (30 out of 167) compared to 11.3% (19 out of 167) in the PCS group (Appendix A). The aOR was 0.34, with a 95% confidence interval of 0.14 to 0.78, indicating a statistically significant reduction in the risk of ESRD in the PCS group.

On the other hand, when evaluating the change in eGFR between the periods before and after the study, there was a marginal significance observed. The mean decrease in eGFR for the control group was −6.8 mL/min/1.73 m^2^, whereas for the PCS group, it was −1.7 mL/min/1.73 m^2^ (*p* value = 0.06).

## 4. Discussion

This study had a specific emphasis on the implementation of PCS in kidney transplantation outpatient clinical settings, with the primary aim of enhancing the clinical outcomes of kidney transplant patients. To the best of our knowledge, this study is the first study to introduce PCS in a Korean kidney transplantation outpatient clinic. As of now, the conceptual framework and extent of pharmaceutical services in Korean outpatient clinics remain unclear and largely unimplemented. We hope that our study will contribute to the successful establishment of outpatient PCS in Korea. Through this endeavor, we anticipate a notable enhancement ultimately leading to an improved patient care.

The provision of PCS by the clinical pharmacist has provided a positive impact on mitigating the decline in kidney function among renal transplant patients through various ways. Notably, polypharmacy has a direct effect on drug-related problems, including drug errors and side effects [20]. On the average, our study participants were prescribed a total of 9.5 medications, thereby significantly increasing the likelihood of encountering drug-related problems. It is noteworthy that the mean eGFR among our subjects were 62.2 mL/min/1.73 m^2^, a value that corresponds to CKD stage 2, making them more vulnerable to adverse events due to reduced drug clearance and increased exposure [21,22,23]. Given these circumstances, the clinical pharmacist has played a pivotal role in proactively adjusting medication dosages and preventing potential medication errors. For instance, immunosuppressive drugs present challenges due to potential drug–drug interactions (DDIs) and significant inter-individual variability [24]. Cases have been found by the clinical pharmacist wherein antiviral medications led to elevated blood levels of tacrolimus or cyclosporine, resulting in acute kidney injury (AKI) during the COVID-19 pandemic. By preemptively identifying and addressing DDIs, the clinical pharmacist has prevented such occurrence. Certain medications such as NSAIDs [25,26] and GI medications (e.g., proton pump inhibitors) [27] could negatively affect patients with renal impairment, necessitating cautious management. In this context, the clinical pharmacist has recommended interventions that prevent patients from using these medications for long-term periods. Moreover, the clinical pharmacist has played a role in managing prevalent complications CKD, such as CKD-MBD [28] and anemia [29].

A potential concern arises because the control group does not require intervention: they might not have DRPs necessitating improvement and could be at a lower risk of renal function deterioration. To address this, we employed propensity score matching, ensuring the baseline characteristics of our two groups were as closely matched as possible, thus evaluating only the intervention’s impact. We found that the protective effect of renal function was greater with increased intervention frequency. In specific subgroups, such as those with diminished renal function, as intervention frequency increased, renal function preservation correspondingly improved: mean declines were −16.2 for a single intervention, −15.5 for 2–3 times, −11.9 for 4–6 times, and −11.3 for 7 or more interventions, illustrating a clear, beneficial trend.

Clinical pharmacist interventions are not only beneficial when a patient has a problem. Their efficacy is amplified when patients actively engage with pharmacists, collaboratively developing a treatment strategy. Some interventions originated from proactive patients seeking the pharmacist’s expertise. We observed clinical improvements when pharmacists introduced symptom-controlling medications or ceased unnecessary medications after discussion with patients. A particularly notable impact on renal protection was seen when discussions covered contingency management such as potential drug side effects, and urinary and respiratory infections, including COVID-19. This aligns with findings showing significant renal protection outcomes from MR and MEM interventions. To summarize, in outpatient contexts, clinical pharmacists can amplify PCS effectiveness by understanding patient preferences and exploring potential unmet needs of clinical improvement.

Indeed, a substantial body of evidence has consistently demonstrated the positive impact of PCS on clinical outcomes in kidney transplant or CKD patients [30,31]. Multiple studies have shown PCS to significantly reduce drug-related problems (in CKD or kidney transplant patients) [17,32,33,34]. Furthermore, PCS has been shown to delay the decline of eGFR in CKD patients [35,36,37], thereby potentially preventing the need for dialysis [38], and mitigating proteinuria [39]. Notably, PCS helped improve patient medication adherence [15], maintain normal hemoglobin levels [40], and regulate blood pressure effectively [41]. These actions have reportedly led to reductions in the incidence of cardiovascular disease [42], hospitalizations, length of hospital stays [43], while enhancing quality of life [44] and reducing costs [43]. In our study as well, many drug-related problems were identified and corrected through the activities of the clinical pharmacist. It is noteworthy that our study demonstrated PCS in an outpatient setting can significantly delay renal function decline and reduce ESRD in kidney transplant recipients. In general, outpatient patients may have less severe diseases than hospitalized patients. However, we were able to prove that PCSs are effective even in outpatient settings. Based on these results, we anticipate that PCSs will be continuously operated in various clinical settings, including hospitalizations and outpatients, and the roles of clinical pharmacists would be expanded. Further research is needed in more diverse patient populations.

This study had several limitations. First, this study was conducted in a single-center, retrospective cohort design, and thus lacked external validity. One should be aware that despite reports suggesting that PCS effectively delays the deterioration of kidney function, there are also cases where the efficacy of PCS has not been demonstrated. In some retrospective studies, the significance of the effects of PCS in kidney transplant patients could not be proven [14,45], and even many randomized controlled trials have reported no discernible effect of PCS in patients with CKD or kidney transplants [33,37,39,46]. Because this study was analyzed retrospectively without randomization, it may not be accurate to attribute the results solely to PCS. Propensity score matching was performed to minimize confounding variables, including patient conditions. However, it was impossible to completely control for the involvement of medical staff (e.g., nurse) other than pharmacists, or decisions related to the physician’s treatment plan. Therefore, caution is required when interpreting our results. Additionally, due to the relatively brief duration of the outpatient treatment period in which PCSs were implemented in our study, there existed challenges in conducting an exhaustive review of medical charts to thoroughly identify the diverse needs of both patients and medical staff. Consequently, there is a possibility that certain drug-related problems may have gone undetected, potentially leading to an underestimation of the advantageous impact of PCSs. To address these limitations and provide a more comprehensive understanding of the impact of PCSs, it is imperative that future research endeavors adopt a multicenter design and encompass a substantial cohort of patients. Through these efforts, a more complete and accurate assessment of the potential advantages of PCSs can be achieved.

## 5. Conclusions

The comprehensive description and positive clinical outcomes demonstrated in our study regarding PCS implementation in a kidney transplantation outpatient clinic hold great promise for wider and efficient provision of PCSs. We envision that the successful application of PCS in this setting will have far-reaching implications, benefiting patients with various chronic diseases, including kidney transplants.

## Figures and Tables

**Figure 1 healthcare-11-02597-f001:**
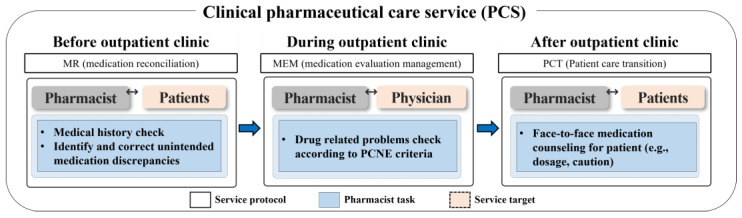
Overview of pharmaceutical care services (PCS). The PCS (medication reconciliation (MR), medication evaluation and management (MEM), and pharmaceutical care transition (PCT) service) was provided to the patients for 3 years in a kidney transplant outpatient clinic.

**Figure 2 healthcare-11-02597-f002:**
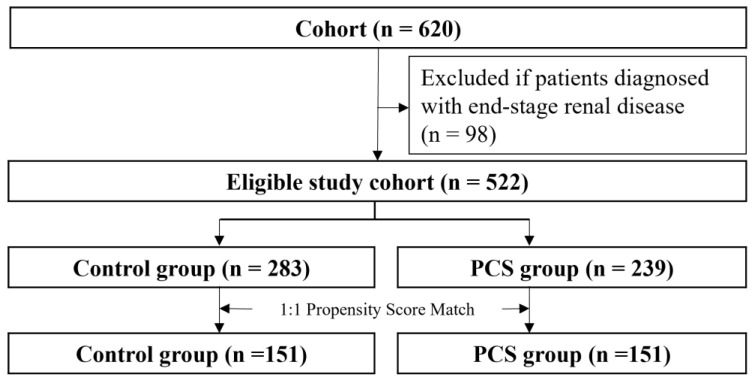
Study flow chart pharmaceutical care services, PCS.

**Figure 3 healthcare-11-02597-f003:**
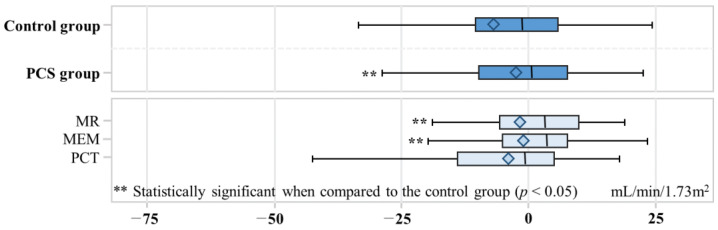
Box plot for change of estimated glomerular filtration rate. The box represents the interquartile range (IQR) and contains the middle 50% of the data. The left edge of the box corresponds to Q1, and the right edge corresponds to Q3. The median is represented as a horizontal line inside the box. The diamond inside the box of a box plot represents the mean of estimated glomerular filtration rate change. The left whisker extends from Q1 to the smallest data point within 1.5 times the IQR, and the right whisker extends from Q3 to the largest data point within 1.5 times the IQR. Medication evaluation and management, MEM; medication reconciliation, MR; pharmaceutical care services, PCS; pharmaceutical care transition, PCT.

**Table 1 healthcare-11-02597-t001:** Baseline characteristics.

	Pre-Match			Post-Match		
Variables	Control n = 283	PCS n = 239	STD	Control n = 151	PCS n = 151	STD
Sex (male)	173 (61.1)	136 (56.9)	0.06	91 (60.3)	83 (55)	0.1
Age (years)	53.8 ± 14.4	54.0 ± 14.0	0.01	53.0 ± 14.4	54.5 ± 13.9	0.1
Diagnosis						
Kidney transplantation	193 (68.2)	205 (85.8)	0.4	126 (83.4)	124 (82.1)	0.04
Chronic kidney disease	74 (26.2)	60 (25.1)	−0.02	39 (25.8)	41 (27.2)	0.03
Kidney donor	40 (14.1)	10 (4.2)	−0.4	6 (4)	8 (5.3)	0.06
Hypertension	159 (56.2)	184 (77.0)	0.5	101 (66.9)	108 (71.5)	0.1
Diabetes Mellitus	75 (26.5)	104 (43.5)	0.4	48 (31.8)	55 (36.4)	0.09
Dyslipidemia	176 (68.2)	175 (73.2)	0.2	109 (72.2)	111 (73.5)	0.03
Number of outpatient visits	6.9 ± 3.5	10.4 ± 5.3	0.8	8.4 ± 3.3	8.2 ± 2.6	−0.1
Number of medications	7.4 ± 7.1	13.3 ± 10.4	0.7	9.4 ± 8.0	9.5 ± 6.5	0.02
Medications						
Diuretics	31 (11.0)	71 (29.7)	0.5	19 (12.6)	25 (16.6)	0.1
Potassium-lowering agents	5 (1.8)	14 (5.9)	0.2	5 (3.3)	2 (1.3)	−0.1
Alkalizing agents	11 (3.9)	31 (13.0)	0.3	8 (5.3)	6 (4.0)	−0.1
Antibiotics	82 (29.0)	116 (48.5)	0.4	55 (36.4)	53 (35.1)	−0.03
Immunosuppressants	196 (69.3)	205 (85.8)	0.4	127 (84.4)	124 (82.1)	−0.05
CKD_MBD medications	34 (12.0)	59 (24.7)	0.3	25 (16.6)	26 (17.2)	0.02
Laboratory findings						
Serum creatinine (mg/dL)	1.29 ± 0.7	1.5 ± 1.0	0.3	1.3 ± 0.8	1.3 ± 0.8	−0.003
BUN (mg/dL)	19.39 ± 9.5	24.1± 13.1	0.4	20.4 ± 11.8	20.8 ± 9.7	0.03
eGFR (mL/min/1.73 m^2^)	64.49 ± 21.4	55.9 ± 23.2	−0.4	62.5 ± 21.5	62 ± 21.8	−0.03
Calcium (mg/dL)	9.4 ± 0.4	9.3 ± 0.5	−0.2	9.4 ± 0.5	9.3 ± 0.5	−0.07
Cholesterol (mg/dL)	182.1 ± 30.1	181.7 ± 33.5	−0.01	184.7 ± 30.3	184.5 ± 32.4	−0.001
Glucose (mg/dL)	108.1 ± 25.5	113.7 ± 44.0	0.2	111.2 ± 29.7	112.3 ± 44.3	0.03
Hemoglobin (g/dL)	13.6 ± 1.7	13.2± ±0.8	−0.2	13.6 ± 1.9	13.4 ± 1.8	−0.09
Albumin (g/dL)	4.5 ± 0.3	4.3 ± 0.4	−0.3	4.4 ± 0.4	4.4 ± 0.4	−0.1
Total bilirubin (mg/dL)	0.8 ± 0.3	0.8 ± 0.3	0.2	0.8 ± 0.4	0.8 ± 0.3	−0.09
Uric acid (mg/dL)	5.8 ± 1.5	6.1 ± 1.7	0.2	5.8 ± 1.5	5.9 ± 0.6	0.04
Platelet (10^9^/L)	224.2 ± 57.5	221 ± 59.0	−0.06	227.5 ± 61.9	224.5 ± 58.0	−0.01
White Blood Cell (10^9^/L)	7.0 ± 1.9	7.4 ± 2.5	0.2	7.2 ± 1.8	7.1 ± 2.0	−0.06

Values are represented as mean ± standard deviation or number (%); BUN, blood urea nitrogen; CKD_MBD, chronic kidney disease-mineral bone disorder; eGFR, estimated glomerular filtration rate; PCS, pharmaceutical care services; STD, standardized difference.

**Table 2 healthcare-11-02597-t002:** Number of pharmaceutical care service (PCS) according to each type of intervention in the intervened group.

Types of PCS	Number of PCS
MR (Identification and correction of unintended medication discrepancies)	113
MEM (Medication managements for drug-related problem)	186
Indication without prescription	67
Insurance issues	66
Drug information	31
Anti-diabetic medication management	16
Dosage adjustments	6
PCT (Medication education for patients)	177
Drug usage	75
Changes in drugs	73
Lifestyle guidance	14
Side effects	9
Supplements	3
Adjusting medication schedules for procedures (e.g., aspirin)	2
Drug identification	1

**Table 3 healthcare-11-02597-t003:** Odds ratio for End Stage Renal Disease.

Outcomes	Incidence		aOR
	Control	PCS	
End Stage Renal Disease	29/151 (19.2)	16/151 (10.6)	0.51 (0.26–0.96)

Values are represented as number (%); aOR, adjusted odds ratio; PCS, pharmaceutical care services.

## Data Availability

The data presented in this study are available upon request from the corresponding author.

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
