# Peer review of "Clinical Effectiveness of Renal Transplant Outpatient Pharmaceutical Care Services in Korea"

_healthcare, 2023, doi:10.3390/healthcare11182597_

Round 1

Reviewer 1 Report

Dear Editors,

This research paper, which has been submitted to the journal Healthcare, offers significant insights into the clinical effectiveness of pharmaceutical care services (PCS) for Korean kidney transplant patients in an outpatient clinic setting. The study's strengths lie in its clinical relevance, patient-centered approach, pharmacist involvement, detailed interventions, statistical methodology, long-term follow-up, ethical considerations, and potential policy impact.

While the study highlights several positive findings, there are several points to consider. The only weakness lies in the statistical analysis and the confounding factors that I addressed in my following comments.

Major issues

-        While propensity score matching can be a valuable method to address confounding factors, it is essential to acknowledge its limitations. The authors are advised to perform a sensitivity analysis to examine the robustness of the results to violations of the propensity score matching assumptions.

-        Some additional factors that could potentially confound the relationship between PCS and the study outcomes should be addressed. i.e.: comorbidities such as hypertension; socioeconomic status and access to healthcare.

-        The overall English language proficiency is good, however, some parts of the manuscript seem to be AI-generated as detected by GPTZero and ZeroGPT.

Author Response

We were pleased to have an opportunity to revise our manuscript now entitled “Clinical effectiveness of renal transplant outpatient pharmaceutical care services in Korea". In the revised manuscript, we have carefully considered reviewers’ comments and suggestions. As instructed, we have attempted to succinctly explain changes made in reaction to all comments. We reply to each comment in point-by-point fashion. We have color coded revised manuscript as text. The responses to the concerns raised by reviewers are below and are color coded as follows: a) Comments from editors or reviewers are shown as text; b) Our responses are shown as text, table, or figure. 
The reviewers’ comments were constructive overall, and we are appreciative of such constructive feedback on our original submission. After addressing the issues raised, we feel the quality of the paper is much improved.

Reviewer 2 Report

Overall, the manuscript is well-written, and the presentation is clear and easy to follow. The impact of PCS is overlooked, and such a study is highly beneficial. There are several comments that could be addressed to improve the manuscript, as detailed below.

Major: 

A significant concern centers around the control selection. On page 3, lines 120-122, the PCS is determined by the pharmacist and provided if the patient needs it. Control patients, in turn, have not received any intervention. Is this due to an absence of needs among control patients? For instance, do these patients display higher medication adherence or encounter no medication-related issues such as dosage errors? It is confusing why these patients experience worse outcomes, and the rationale for selecting them as controls remains unclear. Further clarification may be necessary regarding the comparison between the control group and PCS group to address any potential biases.

Minor: 

1. The author highlights that the usual outpatient interaction is brief, spanning only 3.7 minutes. To provide a comprehensive understanding, the study could provide an estimated duration it takes for a pharmacist to resolve or provide one PCS, encompassing the pre-clinic, during clinic, and post-clinic phases.

2. Could the authors clarify the number of pharmacists involved in this study?

3. On Page 2, Line 85, kindly provide a definition for "DrugTEAM." Likewise, on Page 9, Line 283, provide a definition for "RCTs."

5. In figure 1, does "After outpatient clinic" mean that the medication counseling is provided virtually?

Author Response

(The authors gave the same response as above.)

Reviewer 3 Report

Many thanks for involving me to review this manuscript entitled “Clinical effectiveness of renal transplant outpatient pharmaceutical care services in Korea” which aimed to evaluate the clinical effectiveness of pharmaceutical care service provided to Korean kidney transplants in an outpatient clinic setting. This is very important study which I believe will enrich the scientific background.

Some comments were raised and need to be addressed:

1.      Abstract:

·        Please note that abstract must be structured, include the headings: Background and aim, methods, results, conclusion

·        Add percentages for “, including medication reconciliation (n=113), medication evaluation and management (n=186), and pharmaceutical care transition (n=177) services”

·        This statement is unclear “The change in estimated glomerular filtration rate showed a significant difference between the control and PCS groups, with a decrease of -7.0 mL/min/1.73m2 in the control 19 group and -2.5 mL/min/1.73m2 in the PCS group (P = 0.03).” you were comparing pre and post? Or control vs PCS ?

·        Regarding “Although it is a retrospective study conducted at a single institution, the PCS might contribute to the care of many inflammatory diseases including kidney transplants” I believe this line must be rewritten to show the clinical impact of the findings comprehensively.

2.      Introduction

·        Please add a paragraph to show the research gap and why this study should be published in an international journal.

·        Add the aim of the study at the end of this section comprehensively.

3.      Methods

Please add more details about the sample recruitment method? Random vs convenient?  Consent form? Inclusion and exclusion criteria?

For recruitment did the clinical pharmacist use paper based forms or electronic? 

4.      Discussion

Please elaborate with the critical discussion of the main findings with the clinical implications of it.

Author Response

(The authors gave the same response as above.)

Round 2

Reviewer 1 Report

Dear Editors,

The authors adequately addressed all the previous concerns.

I don't have any further queries. 

Thank you